# Family Food Environments and Their Association with Primary and Secondary Students’ Food Consumption in Beijing, China: A Cross-Sectional Study

**DOI:** 10.3390/nu14091970

**Published:** 2022-05-08

**Authors:** Rong Zhang, Xiaohui Yu, Yingjie Yu, Dandan Guo, Hairong He, Yao Zhao, Wenli Zhu

**Affiliations:** 1Department of Nutrition and Food Hygiene, School of Public Health, Peking University, Beijing 100191, China; zhangrong@pku.edu.cn; 2Key Laboratory of Reproductive Health, China’s Ministry of Health, Beijing 100191, China; 3Beijing Center for Disease Prevention and Control, Beijing 100013, China; yxh770770@sina.com (X.Y.); yyj.jane.1982@163.com (Y.Y.); aldblm.tm@163.com (D.G.); hehairong@bjcdc.org (H.H.)

**Keywords:** family food environment, food consumption, primary and secondary students, China

## Abstract

Family is the most fundamental and proximal context for children, and children’s eating behavior occurs mostly in the home or together with family members. With this study, we aimed to evaluate the distribution of family food environment dimensions and their relationship with healthy and unhealthy food consumption in primary and secondary students in order to provide evidence in the Chinese context and explore environmental solutions for improved child nutrition. Multi-stage stratified cluster sampling was used to conduct a cross-sectional survey among students in Beijing, China, from September 2020 to June 2021. Family food environment (FFE) was measured by the validated Family Food Environment Questionnaire for Chinese School-age Children, which was self-administered by the children’s caregivers. The students were asked to answer questions about food consumption frequencies in the past 7 days. Binary logistic regression models were used to investigate the relationships between food consumption frequency and FFE, and odds ratios (*ORs*) with 95% confidence intervals (*CI*) were computed for independent variables. Among the study population, 9686 students in grades 3–12 and their caregivers completed the survey. The mean score of FFE was 65.7 (±8.4) out of a total possible score of 100, with 76.6% of students categorized as relatively healthier according to their FFE score (≥ 60). Compared with the reference group, students in primary schools and those cared for principally by parents were more likely to be in a healthy FFE category (*p* < 0.05). Daily consumption of fruits and vegetables was reported by 62.6% and 71.6% of students, respectively, and weekly sugared soft drink consumption was reported by 70.9% of the students. Students with a healthier FFE score (≥60) were more likely to consume fruits (*OR* = 1.578, 95% *CI*: 1.428~1.744) and vegetables (*OR* = 1.402, 95% *CI*: 1.263~1.556) but less likely to consume sugared soft drinks (*OR* = 0.592, 95% *CI*: 0.526~0.667). Family food availability (*β* = 0.137), caregivers’ nutritional literacy (*β* = 0.093), meal practices (*β* = 0.079) and food rules (*β* = 0.050) were positively correlated with food consumption behavior (*p* < 0.05). The students with healthier FFE scores (*OR* = 1.130, 95% *CI*: 1.014~1.258) and whose caregiver was obese (*OR* = 2.278, 95% *CI*: 1.973~2.631) were more likely to be overweight. The family food environment plays an important role in shaping food consumption in children. Provision of healthy foods instead of unhealthy foods, positive meal practices and food rules, and nutrition education for parents can promote healthy eating in children.

## 1. Introduction

Presently, children and adolescents are facing double burdens of malnutrition, along with a failure to consume a healthy diet, such as insufficient intake of fruits and vegetables, whole grains, legumes and seafood, as well as considerable exposure to high-fat and high-sugar foods [1,2,3,4]. Random-effects meta-analysis of the Global School-based Student Health Survey between 2008 and 2015 indicated that overall, 34.5% and 20.6% of school-going adolescents consumed fruit and vegetables, respectively, less than once per day; 42.8% drank carbonated soft drinks at least once per day; and 46.1% consumed fast food at least once per week [5]. The pooled national estimates of daily sugar-sweetened beverage consumption in children aged 2 to 18 years from countries in regions particularly burdened by dietary-related chronic illnesses was 326.0 mL (95% confidence interval (*CI*): 288.3~363.8), and the highest estimate was 710.0 mL (95% *CI*: 698.8~721.2) in China [6]. Since the 1980s, Chinese children and adolescents have experienced a nutritional transition from undernutrition to double burden, with a rapid increase in overweight prevalence from 1.1% (1985) to 20.5% (2014) [7]. Meanwhile, eating behavior problems and unbalanced food consumption worsened, with insufficient consumption of healthy foods and excess intake of unhealthy foods [8,9]. Besides nutrition education focusing on individuals, environmental improvement strategies are receiving increased attention.

Children’s food choice is a complex outcome in relation to individual characteristics such as genetics, early-life factors, cognition and emotion, and environmental factors [10]. Food environments range from those that are most proximal to the child’s experience, such as the family, to those that are more distal, such as policies in place at the government and industry level that indirectly influence children [11,12,13]. According to the framework for the Analysis Grid for Elements Linked to Obesity (ANGELO) [14] and an ecological approach [15], food environments exist on two scales—micro settings (home, school and neighborhood) and macro sectors (government, industry and society)—with four types—physical (available), economic (economic accessible), policy (formal or informal rules) and sociocultural (attitudes, beliefs, perceptions and values) environments. Overall, food environments influence the opportunities for healthy eating by interacting with individual factors, which have been demonstrated as effective approaches in improving children’s diet quality and chronic disease risk [2,3,15,16,17].

Among environmental factors, family is the most fundamental and proximal context for children, whose eating behavior occurs mostly in the home or together with family members [18]. Studies have shown family physical and sociocultural environmental factors shape children’s early experiences with food and eating, including food availability, household food expenditure, food rules, mealtimes, and parenting patterns and behavior modelling [19,20,21,22,23,24,25,26,27,28,29]. Systematic reviews have reported that the availability of fruits and vegetables (FV) in the family affect the intake of FV in children [26,27]. Furthermore, the family involvement was associated with adiposity in childhood [21,24].

Observational studies give clues to understand the relationship between family factors and children’s food consumption, but few intervention studies have demonstrated causality effects. Cross-sectional studies have shown that parenting practices related to FV consumption are positively related to adolescents’ FV consumption, but parental limits on junk food/sugar-sweetened beverages (SSBs) are positively related to adolescents’ junk foods and SSBs consumption [28] through autonomous motivation and perceived parental attitudes [29]. Several randomized controlled trials (RCTs) showed that a combination of school-based and family-based interventions could effectively reduce SSB consumption among Chinese school children [30], and family meals interventions significantly reduced SSB consumption among children [31]. However, the Cochrane’s intervention review of 2019 showed that current intervention evidence (RCTs and quasi-RCTs) was insufficient to support the inclusion of caregiver involvement to improve children’s dietary intake behavior and that the quality of the evidence was adversely impacted by the small number of studies with available data, limited effective sample sizes, risk of bias, and imprecision [20]. Additional studies measuring clinically important outcomes using valid and reliable measures, employing appropriate design and power, and following established reporting guidelines are needed [20]. The interactions of multiple environments (e.g., home, school, and neighborhoods) involving children warrant further research.

The association between family environment and children’s eating behaviors is well established, but a multidimensional approach is insufficient, especially in the Chinese context, in view of sociocultural differences. On the other hand, because of the lack of a consistent conceptual framework and assessment instruments, there has been insufficient systematic evaluation of family food environments in children. Previously, we developed and validated the Family Food Environment Questionnaire for Chinese School-age Children [32]. With this study, we aimed to evaluate the family food environment dimensions and their relations with healthy and unhealthy food consumption in primary and secondary students in order to provide evidence in the Chinese context and explore environmental solutions for improved child nutrition.

## 2. Materials and Methods

### 2.1. Study Design and Sampling

The cross-sectional study was conducted from September 2020 to June 2021 in Beijing, China. The target population was primary and secondary students in grades 3~12 in Beijing; 1st~2nd grade students were not involved due to their cognitive disparity. The study participants were selected using a multi-stage stratified cluster sampling strategy to make the sample representative. Three stages were included during the sampling process of students in grades 3~9: (1) Selection of the sample communities: two communities, including one neighborhood and one town (or two neighborhoods if there was no town), were selected separately in 16 central urban and suburban districts of Beijing in terms of geographical location and economic development level of the communities. (2) Selection of the sample schools: two representative schools, including one primary and one junior high school, were selected separately in the above communities in terms of school conditions. (3) Selection of the sample classes: one class was chosen randomly from each grade (3~9) in the above schools. Because there were no senior high schools located in towns, the selection of 10th~12th grade students consisted of two stages: stage 2 and 3. All students (n = 10,000) in the sample class were invited to the survey, as shown in Figure 1.

The study protocol was explained to the subject candidates and their caregivers in a parent–teacher meeting interview. Ultimately, informed written consent was voluntarily obtained from 9912 child–caregiver pairs, and the response rate was 99.1%. The sample is sufficient to represent students in Beijing, China.

The study was approved by the Institutional Review Board of the Beijing Center for Disease Prevention and Control (Beijing, China, approval number 2020-29) and conducted according to the Declaration of Helsinki. The privacy of participant child–caregiver pairs and the confidentiality of their personal information was protected.

### 2.2. Family Food Environment Assessment and Demographic Measurement

Previously, the Family Food Environment Questionnaire for Chinese School-age Children (FFEQ-SC) was developed and validated by our team members. To the best of our knowledge, the FFEQ-SC is the first well-established comprehensive measurement instrument of family food environment (FFE) for Chinese children. The internal reliability of the questionnaire (Cronbach’s α coefficient) was 0.78. The cumulative variance contribution rate of factors was 62.33%, and the goodness of fit index was 0.88 [32]. The FFEQ-SC was self-administered by the children’s caregivers at home, and the investigators explained the queries online via email, telephone or text message.

The conceptual framework of the FFEQ-SC was constructed with reference to the Analysis Grid for Elements Linked to Obesity (ANGELO) [14,15], including physical, economic, policy and sociocultural environments, which comprise of six dimensions: (1) Family socioeconomic status (SES), including family food expenditure and economic status. Family affluence status was assessed using the adjusted “family affluence scale (FAS)”, which has been proven to be a reliable and valid measure of family economic level for Chinese children [33]. Considering the participant family characteristics, three items of FAS were retained: “Does your family own a car, van or truck?”; “Do you have your own bedroom for yourself?”; and “How many times did your family travel for a holiday/vacation last year?”. Despite the fact that family travel may have been affected by the COVID-19 pandemic, the item was retained in view of all participants living in the same context. The item, “How many computers does your family own?” was removed because of the popularization of computers in Chinese families. (2) Family food availability (FA), including the availability of healthy foods (fruits, vegetables, dairy products and coarse food grain) and unhealthy foods (foods high in sugar, fat and salt, as well as sugar-sweetened beverages). (3) Family feeding patterns (FP), including permission (I permit my child to eat what he/she wants, including high-caloric fast food and sweets), restriction (I have to be sure that my child does not eat too many foods high in sugar, fat and salt), enforcement (I enforce my child to eat up the foods. If my child says “I am not hungry” I try to get him/her to eat anyway), role modeling and encouragement patterns (I try to eat a balanced diet in the presence of my child, I encourage my child to eat fruits and vegetables). (4) Family food rules (FR), including dedicated eating (No eating while watching TV/video), food limitation (I limit my child to eating foods high in sugar, fat and salt) and intake request, as well as participation in food preparation activities (My child must help prepare food and do the dishes). (5) Family meal practices (MP), including frequency, location, length and family members present for meals. (6) Caregiver’s nutrition literacy (CNL), including parents’ and other caregivers’ nutrition-related knowledge and skills (nutrition and health, dietary guidelines, food labelling and food portions), as well as discussing nutrition information with children.

Demographic data on the students’ grade and gender, caregivers’ education and family structure were also collected with the questionnaire.

### 2.3. Food Consumption Investigation

Targeted at potential unhealthy eating behaviors, the consumption frequencies of foods were investigated with a self-administered questionnaire. The students were asked to answer the following questions: “During the past 7 days, how many days did you eat: (1) Whole grains; (2) Fruits (do not count fruit juice); (3) Vegetables; (4) Vitamin A-rich vegetables (dark color vegetables); (5) Dairy products (such as milk, yogurt, powder, cheese); (6) Legumes (such as soybean milk, tofu bean curd, dried tofu); (7) Fungi and algae; (8) Fish; (9) Liver meat; (10) Sugared soft drinks; (11) Fried food; (12) Western fast food (characterized by high-fat and high-calorie foods such as burgers, French fries, fried chicken); (13) Breakfast; (14) Snacks (foods intake between continuous formal meals)”. The variety of foods eaten during the past 24 h was also elicited to determine food diversity. Completion of the questionnaire was guided by the investigators in the classroom.

Frequency of consumption was reported using the following categorical responses: “None”, “1–2 days”, “3–4 days”, “5–6 days” and “Everyday”. The items and responses were selected according to the literature [34,35].

According to the Chinese dietary guidelines (2016), the food consumption frequency was recorded and dichotomized into “daily” and “≤6 days” for whole grains, fruits, vegetables, vitamin-A-rich vegetables, dairy products, legumes, breakfast and snacks; and “weekly (at least once a week)” and “none” for fungi and algae, fish, liver, sugared soft drinks, fried food and fast food. Food variety was dichotomized into “≥12” and “<12” on a daily basis. Furthermore, food consumption frequencies were scored from 0 to 2 according to their logistic relations with health outcome. Finally, a composite score was calculated based on food variety and consumption of 13 foods to assess overall food consumption behavior. A higher score indicates healthier food consumption behaviors. Snack consumption was not included in the overall score because its relations with health remain bidirectional.

### 2.4. Anthropometric Data

In Beijing, all students are periodically physically examined each year, and the information is uploaded to the “Beijing School Health Information Management System”. The latest anthropometric data (height and weight) from March to June 2021 were acquired, as permitted by the sample schools and participants.

Body mass index (BMI) was calculated as weight in kilograms divided by the square of the height in meters (kg/m^2^), and the weight status of students was assessed according to Chinese standards of “screening for overweight and obesity among school-age children and adolescents (WS/T 586-2018)” and “screening standard for malnutrition of school-age children and adolescents (WS/T 456-2014)”.

The anthropometric data were imported to the survey database by unique student number.

### 2.5. Variable Value Assignment and Statistical Analysis

Data arrangement and statistical analyses were conducted using EpiData (version 3.1, The EpiData Association, Odense, Denmark) and SPSS (version 27.0, IBM Corp, Armonk, NY, USA).

The FFEQ-SC items were scored according to their logistic relations with diet and health outcome, and the items positively related with health were assigned a positive score, and vice-versa. A composite score was calculated for each respondent based on his or her answers to all 49 items, ranging from 0 to 100. The total score of SES, FA, FP, FR, MP and CNL dimensions ranged from 0 to 8, 16, 18, 24, 17 and 17, respectively. For FFE and its six dimensions, a higher score indicates healthier environments. There is not recognized cut-off, and the study categorized healthy FFE in terms of a composite score if more than 60. Family sociodemographic characteristics (SDCs), such as principal caregivers and their educational level, as well as the number of children in the family, are components of the family environment; these variables were analyzed individually in view of their inconsistent relations with diet and health.

Descriptive statistics were used to present the characteristics of the participants and the distribution of FFE variables. Differences in FFE score were compared among sociodemographic characteristics and food consumption dichotomization using independent-sample *t*-test and one-way ANOVA. Differences in percentage of healthy FFE were compared using chi-square test. Binary logistic regression models were used to investigate the relationships between food consumption frequency and FFE, as well as the relationships between overweight, food consumption and FFE. Odds ratios (*ORs*) with 95% confidence intervals (*CI*) were computed for independent variables. Multiple linear regression was used to explore the relations between overall food consumption behavior score and FFE. The statistical significance level was set at 0.05.

## 3. Results

### 3.1. Characteristics of Participants and Family Food Environment

Among the study population, 9686 students and their caregivers completed the survey, including 35.5% from central urban districts and 64.5% from suburban districts. The percentage of students in primary, junior and senior high school was 42.2%, 29.7% and 28.1%, respectively, and 49.5% were female. In approximately two-thirds (63.3%) of the families, there was only one child. Most (98.1%) of the students were cared for principally by their parents, and one-third of the caregivers reported having a college education or above. More details can be found in Table 1.

The mean family food environment (FFE) score was 65.7 (±8.4) out of a total possible score of 100, ranging from 34.5 to 92.0. A proportion of 76.6% of households had a relatively healthy FFE (≥ 60). The average scores of the six FFE dimensions—family economic status (SES), family food availability (FA), family feeding pattern (FP), family food rules (FR), family meal practices (MP) and caregiver’s nutritional literacy (CNL)—were 4.3 (±1.3) out of a total possible score of 8, 11.4 (±2.2) out of a total possible score of 16, 8.7 (±1.9) out of a total possible score of 18, 16.2(±4.5) out of a total possible score of 24, 12.3 (±1.9) out of a total possible score of 17 and 12.8 (±2.9) out of a total possible score of 17, respectively. The FP score was relatively low, and FA, MP and CNL were assigned higher scores, with average scores of more than seventy percent of the total possible score.

The overall FFE score was significantly different depending on grade, district, caregiver’s education and household income. The FFE score was negatively associated with students’ grade, and primary students had the highest FFE mean score (67.7 ± 7.9), with the lowest score in senior high school students (62.8 ± 8.4, *p* < 0.05). Students in urban schools had a higher FFE score than those in suburban schools (*p* < 0.05). Furthermore, the FFE score was positively correlated with caregivers’ educational level and household income (*p* < 0.05). Although the principal caregiver had no significant relation to overall FFE score, parents who were principal caregivers had higher nutritional literacy and food availability than grandparents (*p* < 0.05), and students with grandparents as caregivers were less likely to have a healthy FFE score (≥ 60), with an *OR* of 0.687 (95% *CI*: 0.489~0.964). The results are listed in Table 1.

### 3.2. Correlations of Family Food Environment Dimensions

Table 2 indicates that the dimensions of FFE were significantly correlated with each other (*p* < 0.05), except that feeding patterns were not correlated with socioeconomic status (SES) or family food rules (FR). The partial correlation coefficients were apparently higher between FA and other dimensions of FFE, ranging from 0.142 (FP) to 0.339 (CNL). The correlation was also strong between MP and FR (r = 0.309).

### 3.3. Family Food Environment in Relation to Students’ Food Consumption

Among student participants, the percentage of daily consumption of whole grains, fruits, vegetables, dairy products and legumes was 27.2%, 62.6%, 71.6%, 54.9% and 13.2%, respectively, and the percentage of weekly consumption of fish, liver meat, sugared soft drinks, fried food and fast food was 72.7%, 36.1%, 70.9%, 67.4% and 52.8%, respectively. Nearly eighty percent had breakfast within the past 7 days, and one-fifth ate more than 12 kinds of foods on the day before completing the questionnaire. More details can be found in Table 3 and Table 4.

A significant association was found between food consumptions and FFE. FFE score was positively related to consumption frequency of whole grains, fruits, vegetables, dairy products, legumes, fish and breakfast but negatively correlated with consumption of sugared soft drinks, fried food, fast food and snacks (*p* < 0.05). Compared with those with a FFE score < 60, for students with a FFE score ≥ 60, the *OR* of daily consumption of whole grains, fruits, vegetables and dairy products was 1.282 (95% *CI*: 1.146~1.435), 1.685 (95% *CI*: 1.527~1.860), 1.466 (95% *CI*: 1.322~1.626) and 1.347 (95% *CI*: 1.223~1.483), respectively, whereas that of weekly consumption of fish, breakfast, sugared soft drinks, fried food and fast food was 1.274 (95% *CI*: 1.147~1.416), 1.895 (95% *CI*: 1.698~2.115), 0.582 (95% *CI*: 0.517~0.654), 0.723 (95% *CI*: 0.649~0.806) and 0.738 (95% *CI*: 0.670~0.814), respectively. Students with a healthier FFE score were more likely to consume diverse foods (*OR* = 1.255, 95% *CI*: 1.110~1.419).

### 3.4. Multiple Regression Analysis Predicting Students’ Food Consumption

Binary logistic regression analysis (Table 5) indicated that a healthier FFE score was a significant predictor of students’ fruit, vegetable and sugared soft drink consumption after adjusting for individual (grade and gender) and household demographic characteristics (district, number of children in family, principal caregiver and their educational level and household income). When the students with a low FFE score (<60) were used as a reference group, the analysis showed that the students with healthier FFE scores (≥ 60) were more likely to consume fruits (*OR* = 1.578, 95% *CI*: 1.428~1.744) and vegetables (*OR* = 1.402, 95% *CI*: 1.263~1.556) but less likely to consume sugared soft drinks (*OR* = 0.592, 95% *CI*: 0.526~0.667).

Multiple linear regression analysis (Table 6) indicated that a healthier FFE score was a significant predictor of healthier overall food consumption behaviors among students after adjusting for individual and household demographic characteristics (*p* < 0.05). When composite FFE score was used as an independent variable (model 1), it was positively associated with food consumption score (*β* = 0.226, *p* < 0.05). When the six dimensions of FFE were used as independent variables (model 2), most of them were positively associated with the food consumption score; the standardized regression coefficients, *β*, were 0.030 for SES, 0.137 for FA, 0.050 for FR, 0.079 for MP and 0.093 for CNL (*p* < 0.05). The results show that health-oriented family food availability, food rules, meal practices and an increase in caregivers’ nutrition literacy could improve food consumption behaviors of students.

### 3.5. Overweight in Relation to Family Food Environment

Among subject students, the overweight prevalence was 40.8%, including 17.4% with pre-obesity and 23.4% with obesity. To analyze the association of FFE and food consumption behaviors with overweight, a binary logistic regression analysis (as shown in Table 7) was conducted, adjusting for individual (grade, gender) and household demographic characteristics (district, number of children in family, principal caregiver and their educational level and household income). The students consuming whole grains daily were less likely to be overweight (*OR* = 0.846, 95% *CI*: 0.761~0.941). Surprisingly, compared with the reference group of non-consumption, consuming fast food weekly was related to lower overweight risk (*OR* = 0.905, 95% *CI*: 0.822~0.997). The most striking result to emerge from the data was that the students with a healthier FFE (score ≥ 60) were more likely to be overweight (*OR* = 1.130, 95% *CI*: 1.014~1.258). Students with an obese caregiver were more likely to be overweight (*OR* = 2.278, 95% *CI*: 1.973~2.631). More details can be found in Table 7.

## 4. Discussion

The results of a representative cross-sectional survey of 3rd to 12th grade students in Beijing, China, showed that the mean FFE was 65.7 (±8.4) out of a total possible score of 100, and 76.6% of households had a relatively healthy FFE (≥ 60). Compared with reference group, students in primary schools cared for principally by parents and whose caregiver had a higher educational level were more likely to be in a healthy FFE. Meanwhile, students with a healthier FFE were more likely to consume healthy foods, including whole grains, fruits and vegetables, but less likely to consume unhealthy foods, such as sugared soft drinks, fried and fast food. In particular, family food availability, caregivers’ nutritional literacy, meal practices and food rules were positively correlated with food consumption behavior.

Family is the most proximal environment for children, and an increasing amount of scientific evidence is focused on the effects of family environment on food consumption behaviors of children. However, the lack of recognized measures of family environment leads to incomparable results. The FFEQ-SC for Chinese children was developed and validated on the basis of ANGELO, [32], which was used to assess physical (FA and MP), economic (SES), policy (FR) and sociocultural environments (FP and CNL) in the present study. This study represents the first comprehensive measurement of FFE in the Chinese context, especially for children. It is necessary to address the conceptual framework and develop more comprehensive assessment instruments guided by recognized theoretical models [36,37,38]. Moreover, the currently available measures of the family environment did not necessarily translate to specific subpopulations in different social circumstances; therefore more testing of some of the identified measures in different population groups is also warranted [38].

Despite the fact that the results of FFE measurement cannot be compared directly, the trends and relations should be discussed. Our study indicated that primary students (middle childhood) had a much healthier FFE compared with secondary students, which might be related to unbalanced attention of families on adolescents. Once children enter secondary school, families may transfer attention to learning and academic concerns instead of eating, which is a focus mainly during early childhood, especially in the Chinese context [39,40].

The six dimensions of FFE were scored differently, and the average FP score was relatively lower, with the average score being 8.7 out of a total possible score of 18. Feeding/parenting patterns are considered a cluster of attitudes and behaviors that extend across multiple contexts of social interaction meant to socialize children, including permission, restriction and enforcement, role modeling and encouragement patterns [12]. Studies have shown that parental modeling and active guidance have the strongest associations with healthy eating behaviors, whereas other patterns, including pressure to eat and restrictive feeding practices, might deprive children of autonomous motivation and self-regulation with regard to eating and were more likely to be associated with picky eating and eating disorders [12,19,22,26,28,29]. In China, parents force children to eat or not to eat some foods (authoritarian pattern), but grandparents tend to permit children to eat what they want (indulgent pattern). A retrospective survey suggested that traditional Chinese feeding habits had significant effects on the occurrence of eating disorders through the synergistic effect of biopsychosocial factors [41]. Interestingly, a systematic review showed that the efficacy of some parenting practices might be dependent on the food consumption context and the age of the child; encouragement might be more effective for healthy foods (*r* = 0.15), whereas restriction is effective for unhealthy foods, especially for children 7 and older (*r* = −0.20); and for children 6 and younger, rewarding with verbal praise could be more effective [26]. The complex and bidirectional effects of different feeding patterns could explain the result (as shown in Table 6) that most FFE dimensions were positively associated with overall food consumption score, except FP score (*β* = −0.009, *p* = 0.343). In our study, feeding patterns were measured as a score but not recognized as different patterns, which should be further explored to measure the relations of different parenting patterns with food consumption.

The present study indicates that in general, the food consumption of students does not adhere to the dietary guidelines, and one-fourth (27.2%) of students consumed whole grains daily; 62.6%, 71.6% and 54.9% consumed fruits, vegetables and dairy products daily, respectively; whereas 70.9% consumed sugared soft drinks weekly. Family has an active role in establishing and promoting behaviors of children that will persist throughout his or her life [11]. Provision of healthy foods instead of unhealthy foods, serving small portions of foods, frequent family meals and positive role modeling by parents could provide children with opportunities to develop self-regulation in eating behaviors, promote healthy eating and favor increased consumption of healthy foods [18,19,25,26,27,42]. The results of our study suggest a significant association between FFE and food consumption; the healthier FFE was, the more frequent the consumption of fruits and vegetables and the less frequent the consumption of sugared soft drinks among students. The overall food consumption score was positively correlated with family food availability, meal practices and food rules. These conclusions are consistent with those other studies [25,28,29,30,31,42,43,44,45,46]. A cohort study of 699 child–parent pairs showed that fruit and vegetable consumption of children was associated with parental encouragement/modeling style (β = 0.68) and unhealthy food availability (β = −0.27); high-calorie beverages were positively correlated with permissive feeding style and unhealthy food availability; and overall diet quality score was positively correlated with healthy food availability, food rules and permissive feeding style [44]. As a family-focused randomized controlled trial with a theoretically driven intervention program for the whole family, the HOME Plus intervention significantly improved parental self-efficacy in identifying appropriate portion sizes and reduced children’s consumption of SSBs [31]. A narrative review on the roles of family in influencing children’s eating behaviors in China showed positive feeding styles were positively correlated with the healthy eating behaviors of children [25]. Some studies do not support these conclusions and showed family interventions had no or little impact on home accessibility, with a smaller impact on consumption [47]. Similarly, the Cochrane 2019 intervention review concluded that current evidence is insufficient to support the inclusion that caregiver involvement in interventions improves children’s dietary intake [20]. More high-quality studies are needed.

Parents’ nutritional literacy is another important sociocultural dimension affecting self-efficacy and children’s knowledge and skill, as well as behavioral and health status [48]. This study showed that caregivers’ nutritional literacy (CNL) was positively correlated with other dimensions of FFE (*p* < 0.05), especially food availability (*r* = 0.339), which indicated nutrition education of caregivers could improve the family food environment, including food availability, feeding patterns, food rules and meal practices. The results showed that CNL was positively correlated with overall food consumption behaviors of children, and parents had a higher nutritional literacy level than grandparents (*p* < 0.05), so it is preferable for parents to care for their own children. A narrative review showed that parental education levels, health awareness and nutritional knowledge were positively correlated with healthy eating behaviors of Chinese children [25]. However, another review suggested that it was uncertain whether parent nutritional education intervention was effective in increasing fruit and vegetable consumption in children aged five years and under [43].

Finally, the relation of FFE with overweight in children was analyzed briefly in the present study. The most striking result was that the children with a healthier FFE were more likely to be overweight (*OR* = 1.130, 95% *CI*: 1.014~1.258). Bidirectionality in parent–child interactions is likely, as parenting influences child eating and weight, although child eating and weight also influence parenting. For example, if a child is overweight, it is reasonable to expect that parents may adopt a more restrictive feeding practice with the intent of limiting the child’s portion sizes and paying much closer attention to nutrition and health matters, which would improve the food environment. Our study also showed the caregivers’ weight was positively related to children’s weight, and children with an obese caregiver were more likely to be overweight (*OR* = 2.278, 95% *CI*: 1.973~2.631). In view of the complexity of overweight, other studies resulted in different conclusions about family environment and childhood overweight [19,21,22,24,37,44]. An umbrella review supported the inclusion of a parent component in both treatment and prevention interventions to improve child weight/weight status outcomes [21]. Another systematic review in 2020 showed that the home media environment was most consistently associated with adiposity in childhood, although findings for home food environments were less consistent [24].

There are some limitations that should be noted in our study. Firstly, the cross-sectional design could identify associations but could not determine the direction of the association. All survey data were collected from the participants’ self-reports, which may have introduced self-report bias. Secondly, despite the fact that FFEQ-SC was developed as a comprehensive measure to assess family food environment, the questionnaire and conceptualization of FFE are still not universally acknowledged, which hinders the comparison of the results with those of other studies. Thirdly, as a behavioral outcome, food consumption was correlated with not only FFE but also individual nutritional knowledge, which was not investigated and included in the linear regression analysis. Therefore, the model presented in Table 6 was able to explain less than 10% of variances. Similarly, as a health outcome, weight status was related with not only eating behavior but also physical activity (environment), which was not investigated and could influence the relations. Finally, besides FFE, other food environments, including school, neighborhood and expenditure environments, could affect food consumption, but these environments were not investigated and adjusted in the regression analysis.

## 5. Conclusions

Our findings support the assertion that the family food environment plays an important role in shaping food consumption in children. Students with a healthier FFE were more likely to consume healthy foods, including fruits and vegetables, but less likely to consume unhealthy foods, such as sugared soft drinks. Provision of healthy foods instead of unhealthy foods, positive meal practices and food rules, as well as nutritional education of parents could promote healthy eating in children. Robust longitudinal research using comprehensive measures of the holistic food environment, including school and neighborhood, as well as the physical activity environment, is needed to better identify which aspects contribute to healthy eating and weight status in childhood.

## Figures and Tables

**Figure 1 nutrients-14-01970-f001:**
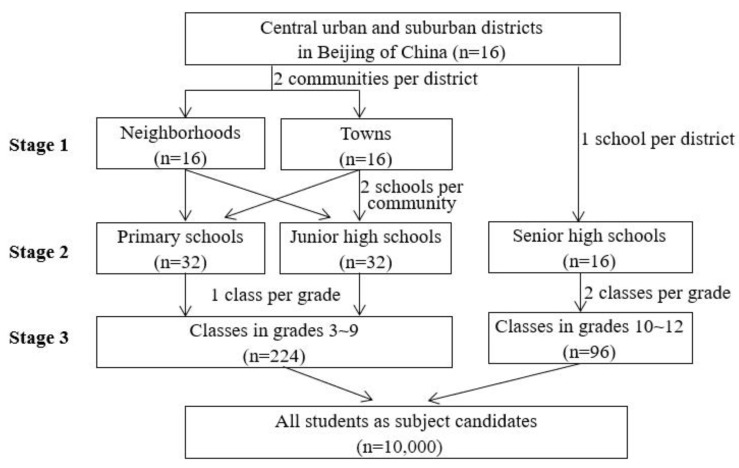
Multi-stage stratified cluster sampling process of the study.

**Table 1 nutrients-14-01970-t001:** Individual and household demographic characteristics of family food environments.

Group	*N* (%)	Family Food Environment Score (Mean ± SD)							Total Score ≥ 60	
		Family Socioeconomic Status	Family Food Availability	Family Feeding Patterns	FAMILY Food Rules	FAMILY Meal Practices	Caregiver’s Nutritional Literacy	Total Score	*n* (%)	*ORs*(95% *CI*)
Total	9686(100.0)	4.3 ± 1.3	11.4 ± 2.2	8.7 ± 1.9	16.2 ± 4.5	12.3 ± 1.9	12.8 ± 2.9	65.7 ± 8.4	7419(76.6)	
Grade										
3~6	4092(42.2)	4.2 ± 1.4 ^a^	11.9 ± 2.0 ^a^	8.9 ± 1.8 ^a^	17.1 ± 3.9 ^a^	12.7 ± 1.8 ^a^	12.9 ± 2.9 ^a^	67.7 ± 7.9 ^a^	3461(84.6)	—
7~9	2875(29.7)	4.3 ± 1.3 ^ab^	11.3 ± 2.2 ^b^	8.6 ± 1.9 ^b^	16.2 ± 4.5 ^b^	12.3 ± 1.9 ^b^	12.7 ± 2.9 ^b^	65.4 ± 8.4 ^b^	2170(75.5)	0.561(0.498~0.633) *
10~12	2719(28.1)	4.3 ± 1.1 ^b^	10.9 ± 2.2 ^c^	8.5 ± 2.0 ^c^	14.7 ± 5.1 ^c^	11.7 ± 2.0 ^c^	12.7 ± 2.8 ^ab^	62.8 ± 8.4 ^c^	1788(65.8)	0.350(0.312~0.393) *
Gender										
Male	4887(50.5)	4.3 ± 1.3 ^a^	11.4 ± 2.2 ^a^	8.7 ± 1.9	16.2 ± 4.5	12.3 ± 1.9	12.7 ± 2.9	65.6 ± 8.5	3706(75.8)	—
Female	4799(49.5)	4.2 ± 1.3 ^b^	11.5 ± 2.1 ^b^	8.7 ± 1.9	16.1 ± 4.5	12.3 ± 1.9	12.8 ± 2.8	65.7 ± 8.4	3713(77.4)	1.090(0.992~1.197)
District										
Urban	3439(35.5)	4.4 ± 1.4 ^a^	11.7 ± 2.1 ^a^	8.9 ± 1.8 ^a^	15.9 ± 4.5 ^a^	12.3 ± 1.9	13.3 ± 2.8 ^a^	66.6 ± 8.4 ^a^	2772(80.6)	—
Suburb	6247(64.5)	4.2 ± 1.3 ^b^	11.3 ± 2.2 ^b^	8.6 ± 1.9 ^b^	16.3 ± 4.6 ^b^	12.3 ± 1.9	12.5 ± 2.9 ^b^	65.2 ± 8.4 ^b^	4647(74.4)	0.699(0.631~0.774) *
Number of children										
1	6133(63.3)	4.3 ± 1.3 ^a^	11.5 ± 2.1	8.7 ± 1.9 ^a^	15.9 ± 4.6 ^a^	12.2 ± 2.0 ^a^	12.9 ± 2.9 ^a^	65.6 ± 8.6	4678(76.3)	—
≥2	3551(36.7)	4.2 ± 1.4 ^b^	11.4 ± 2.2	8.6 ± 1.9 ^b^	16.6 ± 4.4 ^b^	12.5 ± 1.8 ^b^	12.5 ± 2.9 ^b^	65.8 ± 8.2	2740(77.2)	1.051(0.953~1.159)
Principal caregiver										
Parents	9500(98.1)	4.3 ± 1.3 ^a^	11.5 ± 2.2 ^a^	8.7 ± 1.9	16.2 ± 4.5	12.3 ± 1.9	12.8 ± 2.9 ^a^	65.7 ± 8.4	7290(76.7)	—
Grandparents	160 (1.7)	3.8 ± 1.6 ^b^	11.0 ± 2.3 ^b^	8.9 ± 2.2	16.6 ± 4.6	12.2 ± 2.3	12.0 ± 3.1 ^b^	64.4 ± 8.2	111(69.4)	0.687(0.489~0.964) *
Caregiver’s educational level										
≤Junior high school	1603(16.5)	3.8 ± 1.3 ^a^	10.7 ± 2.2 ^a^	8.5 ± 2.0 ^a^	16.1 ± 4.7	12.2 ± 2.0 ^a^	11.6 ± 2.9 ^a^	62.9 ± 8.4 ^a^	1047(65.3)	—
High school	2470(25.5)	4.1 ± 1.3 ^b^	11.1 ± 2.1 ^b^	8.5 ± 2.0 ^a^	16.0 ± 4.7	12.2 ± 1.9 ^a^	12.3 ± 2.8 ^b^	64.3 ± 8.3 ^b^	1789(72.4)	1.395(1.218~1.598) *
Junior college	2383(24.6)	4.3 ± 1.3 ^c^	11.6 ± 2.1 ^c^	8.7 ± 1.8 ^b^	16.4 ± 4.4	12.3 ± 1.9 ^b^	12.8 ± 2.8 ^c^	66.2 ± 8.2 ^c^	1880(78.9)	1.985(1.721~2.288) *
≥College	3230(33.3)	4.7 ± 1.2 ^d^	11.9 ± 2.1 ^d^	8.9 ± 1.8 ^c^	16.2 ± 4.4	12.4 ± 1.9 ^b^	13.7 ± 2.6 ^d^	67.7 ± 8.1 ^d^	2703(83.7)	2.724(2.371~3.130) *
Annual household income per capita (yuan)										
<20,000	1685(17.4)	3.8 ± 1.3 ^a^	10.9 ± 2.2 ^a^	8.6 ± 2.0 ^a^	16.3 ± 4.6	12.1 ± 2.0 ^a^	12.0 ± 2.9 ^a^	63.7 ± 8.4 ^a^	1163(69.0)	—
20,000~39,999	2127(22.0)	4.0 ± 1.3 ^b^	11.3 ± 2.2 ^b^	8.6 ± 1.9 ^ab^	16.2 ± 4.7	12.3 ± 2.0 ^ab^	12.5 ± 2.9 ^b^	64.8 ± 8.4 ^b^	1573(74.0)	1.274(1.106~1.468) *
40,000~69,999	2539(26.2)	4.2 ± 1.3 ^c^	11.5 ± 2.0 ^c^	8.7 ± 1.9 ^bc^	16.1 ± 4.4	12.4 ± 1.8 ^b^	12.9 ± 2.8 ^c^	65.8 ± 8.2 ^c^	1969(77.6)	1.550(1.349~1.782) *
≥70,000	3329(34.4)	4.7 ± 1.3 ^d^	11.8 ± 2.1 ^d^	8.8 ± 1.9 ^c^	16.1 ± 4.5	12.3 ± 1.9 ^b^	13.3 ± 2.8 ^d^	67.1 ± 8.4 ^d^	2708(81.3)	1.957(1.710~2.240) *

a, b, c and d indicate significant differences among groups (*p* < 0.05); *: *p* < 0.05.

**Table 2 nutrients-14-01970-t002:** Partial correlation between family food environment dimensions.

Dimension of Family Food Environment	Family Socioeconomic Status	Family Food Availability	Family Feeding Patterns	Family Food Rules	Family Meal Practices	Caregiver’s Nutritional Literacy
Family food availability	0.061 *					
Family feeding patterns	0.011	0.142 *				
Family food rules	0.029 *	0.276 *	0.002			
Family meal practices	0.032 *	0.283 *	0.054 *	0.309 *		
Caregiver’s nutritional literacy	0.097 *	0.339 *	0.069 *	0.133 *	0.197 *	
Total score	0.237 *	0.633 *	0.306 *	0.734 *	0.556 *	0.585 *

*: *p* < 0.05, adjusted for students’ grade, gender and district.

**Table 3 nutrients-14-01970-t003:** Healthy Food Consumption in Relation to Family Food Environment.

Food Consumption During the Past 7 Days	*N* (%)	Family Food Environment Score (Mean ± SD)							Total Score ≥ 60	
		Family Socioeconomic Status	Family Food Availability	Family Feeding Patterns	Family Food Rules	Family Meal Practices	Caregiver’s Nutritional Literacy	Total Score	*n* (%)	*ORs*(95% *CI*)
Whole grains										
≤6 days	7052(72.8)	4.3 ± 1.3 ^a^	11.4 ± 2.1 ^a^	8.7 ± 1.9	16.1 ± 4.6 ^a^	12.2 ± 1.9 ^a^	12.7 ± 2.9 ^a^	65.3 ± 8.4 ^a^	5305(75.2)	—
Daily	2630(27.2)	4.2 ± 1.4 ^b^	11.7 ± 2.2 ^b^	8.7 ± 2.0	16.5 ± 4.4 ^b^	12.5 ± 1.9 ^b^	13.1 ± 2.9 ^b^	66.7 ± 8.4 ^b^	2111(80.3)	1.282(1.146~1.435) *
Fruits										
≤6 days	3623(37.4)	4.2 ± 1.3 ^a^	11.0 ± 2.1 ^a^	8.6 ± 1.9 ^a^	15.6 ± 4.7 ^a^	12.0 ± 2.0 ^a^	12.4 ± 2.9 ^a^	63.7 ± 8.6 ^a^	2494(68.8)	—
Daily	6063(62.6)	4.3 ± 1.3 ^b^	11.7 ± 2.1 ^b^	8.8 ± 1.9 ^b^	16.5 ± 4.4 ^b^	12.5 ± 1.9 ^b^	13.0 ± 2.8 ^b^	66.8 ± 8.2 ^b^	4925(81.2)	1.685(1.527~1.860) *
Vegetables										
≤6 days	2724(28.1)	4.2 ± 1.3 ^a^	11.0 ± 2.1 ^a^	8.7 ± 1.9	15.8 ± 4.7 ^a^	12.1 ± 1.9 ^a^	12.3 ± 2.9 ^a^	64.1 ± 8.4 ^a^	1937(71.1)	—
Daily	6937(71.6)	4.3 ± 1.3 ^b^	11.6 ± 2.1 ^b^	8.7 ± 1.9	16.3 ± 4.5 ^b^	12.4 ± 1.9 ^b^	13.0 ± 2.8 ^b^	66.2 ± 8.4 ^b^	5461(78.7)	1.466(1.322~1.626) *
Dairy Products										
≤6 days	4360(45.0)	4.2 ± 1.3 ^a^	11.2 ± 2.2 ^a^	8.7 ± 1.9	16.0 ± 4.6 ^a^	12.2 ± 1.9 ^a^	12.5 ± 2.9 ^a^	64.8 ± 8.5 ^a^	3198(73.3)	—
Daily	5322(54.9)	4.3 ± 1.3 ^b^	11.6 ± 2.1 ^b^	8.7 ± 1.9	16.3 ± 4.5 ^b^	12.4 ± 1.9 ^b^	13.0 ± 2.9 ^b^	66.4 ± 8.3 ^b^	4219(79.3)	1.347(1.223~1.483) *
Legumes										
≤6 days	8401(86.7)	4.3 ± 1.3	11.4 ± 2.1 ^a^	8.7 ± 1.9	16.1 ± 4.5 ^a^	12.3 ± 1.9 ^a^	12.8 ± 2.9	65.6 ± 8.5 ^a^	6406(76.3)	—
Daily	1283(13.2)	4.3 ± 1.3	11.6 ± 2.3 ^b^	8.6 ± 2.1	16.6 ± 4.6 ^b^	12.4 ± 1.9 ^b^	12.8 ± 2.9	66.3 ± 8.2 ^b^	1011(78.8)	1.197(1.034~1.385) *
Fish										
None	2641(27.3)	4.1 ± 1.3 ^a^	11.3 ± 2.1 ^a^	8.7 ± 1.9	16.0 ± 4.7 ^a^	12.2 ± 2.0 ^a^	12.5 ± 2.9 ^a^	64.7 ± 8.6 ^a^	1939(73.4)	—
Weekly	7042(72.7)	4.3 ± 1.3 ^b^	11.5 ± 2.2 ^b^	8.7 ± 1.9	16.2 ± 4.5 ^b^	12.3 ± 1.9 ^b^	12.9 ± 2.9 ^b^	66.0 ± 8.3 ^b^	5478(77.8)	1.274(1.147~1.416) *
Liver										
None	6180(63.8)	4.2 ± 1.3 ^a^	11.4 ± 2.1	8.7 ± 1.9 ^a^	16.2 ± 4.5	12.3 ± 1.9	12.7 ± 2.9 ^a^	65.6 ± 8.4	4737(76.7)	—
Weekly	3494(36.1)	4.4 ± 1.3 ^b^	11.5 ± 2.2	8.6 ± 1.9 ^b^	16.2 ± 4.6	12.3 ± 1.9	12.9 ± 2.9 ^b^	65.7 ± 8.5	2675(76.6)	1.147(1.036~1.269) *
Breakfast										
≤6 days	2018(20.8)	4.2 ± 1.3 ^a^	10.7 ± 2.1 ^a^	8.5 ± 2.0 ^a^	15.1 ± 4.9 ^a^	11.7 ± 1.9 ^a^	12.2 ± 3.0 ^a^	62.5 ± 8.4 ^a^	1290(63.9)	—
Daily	7660(79.1)	4.3 ± 1.3 ^b^	11.6 ± 2.1 ^b^	8.7 ± 1.9 ^b^	16.4 ± 4.4 ^b^	12.5 ± 1.9 ^b^	12.9 ± 2.8 ^b^	66.5 ± 8.2 ^b^	6121(79.9)	1.895(1.698~2.115) *
Food variety yesterday										
<12	7620(78.7)	4.2 ± 1.3 ^a^	11.4 ± 2.1	8.7 ± 1.9 ^a^	16.2 ± 4.5	12.3 ± 1.9 ^a^	12.7 ± 2.9 ^a^	65.4 ± 8.4 ^a^	5768(75.7)	—
≥12	2044(21.1)	4.4 ± 1.4 ^b^	11.6 ± 2.2	8.8 ± 1.9 ^b^	16.2 ± 4.6	12.4 ± 1.9 ^b^	13.2 ± 2.8 ^b^	66.7 ± 8.5 ^b^	1635(80.0)	1.255(1.110~1.419) *

a, b indicate significant differences between groups (*p* < 0.05); “weekly” indicates food consumption at least once a week. *: *p* < 0.05, adjusted for students’ grade, gender and district.

**Table 4 nutrients-14-01970-t004:** Unhealthy food and snack consumption in relation to family food environment.

Food Consumption during the Past 7 Days	*N* (%)	Family Food Environment Score (Mean ± SD)							Total Score ≥ 60	
		Family Socioeconomic Status	Family Food Availability	Family Feeding Patterns	Family Food Rules	Family Meal Practices	Caregiver’s Nutritional Literacy	Total Score	*n* (%)	*ORs*(95% *CI*)
Sugared soft drinks										
None	2812(29.0)	4.2 ± 1.4 ^a^	12.1 ± 2.0 ^a^	8.9 ± 1.9 ^a^	17.0 ± 4.3 ^a^	12.6 ± 1.9 ^a^	13.2 ± 2.8 ^a^	67.0 ± 8.5 ^a^	2369(84.2)	—
Weekly	6865(70.9)	4.3 ± 1.3 ^b^	11.2 ± 2.1 ^b^	8.6 ± 1.9 ^b^	15.8 ± 4.6 ^b^	12.2 ± 1.9 ^b^	12.6 ± 2.9 ^b^	64.7 ± 8.3 ^b^	5041(73.4)	0.582(0.517~0.654) *
Fried food										
None	3155(32.6)	4.3 ± 1.4	11.9 ± 2.1 ^a^	8.8 ± 1.9 ^a^	16.7 ± 4.3 ^a^	12.5 ± 1.9 ^a^	13.1 ± 2.9 ^a^	67.3 ± 8.3 ^a^	2581(81.8)	—
Weekly	6526(67.4)	4.3 ± 1.3	11.2 ± 2.2 ^b^	8.6 ± 1.9 ^b^	15.9 ± 4.6 ^b^	12.2 ± 1.9 ^b^	12.6 ± 2.9 ^b^	64.9 ± 8.4 ^b^	4834(74.1)	0.723(0.649~0.806) *
Fast food										
None	4564(47.1)	4.2 ± 1.3 ^a^	11.7 ± 2.1 ^a^	8.8 ± 1.9 ^a^	16.5 ± 4.4 ^a^	12.5 ± 1.9 ^a^	13.0 ± 2.9 ^a^	66.7 ± 8.3 ^a^	3640(79.8)	—
Weekly	5119(52.8)	4.3 ± 1.3 ^b^	11.2 ± 2.2 ^b^	8.6 ± 1.9 ^b^	15.9 ± 4.6 ^b^	12.1 ± 1.9 ^b^	12.6 ± 2.9 ^b^	64.7 ± 8.4 ^b^	3777(73.8)	0.738(0.670~0.814) *
Snacks										
≤6 days	8412(86.8)	4.3 ± 1.3 ^a^	11.5 ± 2.1 ^a^	8.7 ± 1.9 ^a^	16.2 ± 4.5 ^a^	12.3 ± 1.9 ^a^	12.8 ± 2.9	65.8 ± 8.4 ^a^	6464(76.8)	—
Daily	1272(13.1)	4.4 ± 1.3 ^b^	11.3 ± 2.2 ^b^	8.6 ± 2.1 ^b^	15.7 ± 4.6 ^b^	12.1 ± 1.9 ^b^	12.9 ± 2.9	64.9 ± 8.5 ^b^	954(75.0)	0.952(0.828~1.095)

a, b indicate significant differences between groups (*p* < 0.05); “weekly” indicates food consumption at least once a week. *: *p* < 0.05, adjusted for students’ grade, gender and district.

**Table 5 nutrients-14-01970-t005:** Logistic regression analysis of food consumption.

Independent Variable	Daily Fruit Consumption			Daily Vegetable Consumption			Weekly Sugared Soft Drink Consumption		
	*B*	*p*	ORs (95%CI)	*B*	*p*	ORs (95%CI)	*B*	*p*	ORs (95%CI)
constant term	−0.242	0.014		0.227	0.026		1.239	<0.001 *	
Grade									
3~6	—		—	—		—	—		—
7~9	−0.212	<0.001 *	0.809(0.728~0.898)	0.238	<0.001 *	1.269(1.135~1.418)	0.491	<0.001 *	1.634(1.466~1.820)
10~12	−0.809	<0.001 *	0.445(0.400~0.495)	−0.012	0.840	0.989(0.884~1.105)	0.633	<0.001 *	1.884(1.678~2.115)
Gender (Female)	0.302	<0.001 *	1.352(1.241~1.473)	0.257	<0.001 *	1.293(1.182~1.415)	−0.278	<0.001 *	0.757(0.692~0.828)
District (Suburb)	0.209	0.555	1.209(0.935~1.132)	−0.093	0.071	0.911(0.823~1.008)	−0.026	0.614	0.975(0.882~1.077)
Number of children (≥2)	−0.067	0.152	0.935(0.853~1.025)	−0.033	0.506	0.968(0.879~1.065)	0.001	0.989	1.001(0.909~1.101)
Principal caregiver (grandparents)	0.317	0.071	1.373(0.974~1.937)	0.047	0.791	1.048(0.741~1.482)	−0.214	0.220	0.807(0.573~1.137)
Caregiver’s education									
≤Junior high school	—		—	—		—			
High school	0.294	<0.001 *	1.341(1.174~1.532)	0.111	0.116	1.118(0.973~1.284)	−0.004	0.961	0.996(0.857~1.158)
Junior college	0.493	<0.001 *	1.637(1.422~1.885)	0.261	0.001 *	1.298(1.120~1.504)	−0.098	0.214	0.907(0.777~1.058)
≥College	0.608	<0.001 *	1.838(1.591~2.123)	0.417	<0.001 *	1.517(1.303~1.766)	−0.290	<0.001 *	0.748(0.640~0.874)
Household income per capita (yuan)									
<20,000	—		—	—		—			
20,000~39,999	0.128	0.062	1.136(0.993~1.299)	0.182	0.012 *	1.199(1.042~1.381)	−0.035	0.634	0.965(0.834~1.117)
40,000~69,999	0.193	0.004 *	1.213(1.062~1.386)	0.115	0.105	1.122(0.976~1.289)	0.019	0.795	1.019(0.883~1.177)
≥70,000	0.306	<0.001 *	1.358(1.186~1.555)	0.123	0.088	1.131(0.982~1.302)	0.154	0.037 *	1.166(1.009~1.348)
Family food environment score (≥60)	0.456	<0.001 *	1.578(1.428~1.744)	0.338	<0.001 *	1.402(1.263~1.556)	−0.524	<0.001 *	0.592(0.526~0.667)

*: *p* < 0.05; “weekly” indicates food consumption at least once a week.

**Table 6 nutrients-14-01970-t006:** Multiple linear regression analysis of overall food consumption score.

Independent Variable	Model 1			Model 2		
	*B*	*β*	*p*	*B*	*β*	*p*
constant term	10.857		<0.001 *	10.554		<0.001 *
Grade	0.031	0.026	0.013 *	0.029	0.024	0.024 *
Gender	0.087	0.013	0.200	0.061	0.009	0.363
District	−0.222	−0.031	0.003 *	−0.208	−0.029	0.005 *
Number of children	0.041	0.006	0.572	0.037	0.005	0.609
Principal caregiver	−0.050	−0.002	0.810	0.030	0.001	0.886
Caregiver’s education	0.314	0.100	<0.001 *	0.278	0.088	<0.001 *
Household income	0.093	0.030	0.006 *	0.070	0.022	0.043 *
Family food environments	0.092	0.226	<0.001 *	—	—	—
Family socioeconomic status	—	—	—	0.078	0.030	0.003 *
Family food availability	—	—	—	0.219	0.137	<0.001 *
Family feeding pattern	—	—	—	−0.017	−0.009	0.343
Family food rules	—	—	—	0.038	0.050	<0.001 *
Family meal practices	—	—	—	0.141	0.079	<0.001 *
Caregiver’s nutritional literacy	—	—	—	0.111	0.093	<0.001 *
	*F* = 100.273, *p* < 0.05			*F* = 71.574, *p* < 0.05		

Variable values: gender (1: male, 2: female); district (1: urban, 2: suburban); caregiver (1: parent, 2: grandparent, 3: others); caregiver’s education (1: ≤junior high school, 2: high school, 3: junior college, 4: ≥college); household income (1: <20,000, 2: 20,000~39,999, 3: 40,000~69,999, 4: ≥70,000). *: *p* < 0.05.

**Table 7 nutrients-14-01970-t007:** Logistic regression analysis of students’ overweight status.

Independent Variable	*B*	*p*	ORs (95%CI)
constant term	0.024	0.853	
Grade			
3~6	—		—
7~9	−0.122	0.026 *	0.885(0.794~0.986)
10~12	−0.342	<0.001 *	0.710(0.632~0.797)
Gender (Female)	−0.870	<0.001 *	0.419(0.383~0.458)
District (Suburb)	0.299	<0.001 *	1.349(1.223~1.487)
Number of children (≥2)	−0.093	0.051	0.911(0.830~1.000)
Principal caregiver (grandparents)	−0.040	0.822	0.961(0.676~1.364)
Caregiver’s education			
≤Junior high school	—		—
High school	0.078	0.277	1.081(0.939~1.244)
Junior college	0.151	0.044 *	1.163(1.004~1.348)
≥College	0.094	0.223	1.098(0.944~1.278)
Caregiver’s weight status			
Normal	—		—
Overweight	0.455	<0.001 *	1.576(1.429~1.737)
Obese	0.823	<0.001 *	2.278 (1.973~2.631)
Wasted	−0.292	0.024 *	0.747 (0.579~0.963)
Household income per capita			
<20,000	—		—
20,000~39,999	−0.084	0.235	0.919(0.800~1.056)
40,000~69,999	−0.220	0.002 *	0.803(0.699~0.922)
≥70,000	−0.302	<0.001 *	0.739(0.643~0.850)
Family food environments (≥ 60)	0.122	0.026 *	1.130(1.014~1.258)
Food consumption			
Whole grains (daily)	−0.167	0.002 *	0.846(0.761~0.941)
Fruits (daily)	0.085	0.109	1.089(0.981~1.208)
Vegetables (daily)	−0.004	0.943	0.996(0.892~1.112)
Vitamin A-rich vegetables (daily)	−0.003	0.960	0.997(0.902~1.103)
Dairy products (daily)	0.010	0.834	1.010(0.919~1.111)
Legumes (daily)	0.179	0.011 *	1.196(1.042~1.372)
Fungi and algae (weekly)	−0.040	0.488	0.960(0.857~1.077)
Fish (weekly)	−0.023	0.667	0.977(0.879~1.086)
Liver (weekly)	0.017	0.733	1.017(0.923~1.121)
Sugared soft drinks (weekly)	0.038	0.479	1.039(0.935~1.154)
Fried food (weekly)	−0.022	0.683	0.979(0.882~1.086)
Fast food (weekly)	−0.099	0.043 *	0.905(0.822~0.997)
Breakfast (daily)	−0.086	0.137	0.918(0.819~1.028)
Snacks (daily)	−0.276	<0.001 *	0.759(0.662~0.869)
Food variety (≥12)	0.092	0.099	1.096(0.983~1.223)

*: *p* < 0.05; “weekly” indicates food consumption at least once a week.

## Data Availability

The data presented in this study are available on request from the corresponding author. The data are not publicly available due to privacy.

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
