# Peer review of "Family Food Environments and Their Association with Primary and Secondary Students’ Food Consumption in Beijing, China: A Cross-Sectional Study"

_nutrients, 2022, doi:10.3390/nu14091970_

Round 1
Reviewer 1 Report
Thank you for the opportunity to review this article entitled "Family Food Environments and Their Association with Primary and Secondary Students’ Food Consumption in Beijing of China: A Cross-Sectional Study”. Findings from this paper are interesting where gaps have been identified in the influence of family environment on children’s nutritional status in
non-Western culture. Nevertheless, there are issues that are to be addressed in this paper as follows:
1) Abstract
• Replace ‘Objective’ with ‘Background’ to outline the context of the paper before stating the study objective
• Methods: Include a sentence on how results are analysed
• Results (Line 22): Remove ‘finally’
• Results (line 30): Remove ‘Especially’
• Keyword: Suggest to include ‘China’
2) Introduction
• The introduction section can be shortened and reorganised for better flow
o Line 67 to 89: Suggest to combine both paragraph and discuss on family food environment, and make it succinct
o Line 104 to 115: Combine with Paragraph 1 to provide an overview on the state of the problem globally and in China/ Beijing, and make it succinct
• Minor edits as follows:
o Line 40: Replace multiple with double/triple
o Line 50 to 52: Remove the sentence
o Line 53: Remove ‘actually’
o Line 55: Sentence too long, suggest to stop after ‘factors’ and start new sentence.
o Line 63 to 65: Sentence too long and unclear, please reword
o Line 90 to 92: Sentence too long and unclear, please reword
3) Materials and Methods
• Study Design and Sampling, Line 123-124:
o Suggest to remove sentence
• Family Food Environment Assessment and Demographic Measure:
o Line 150: Change to passive voice
o Line 163: Define ‘SDS’
o Line 185 to 186: Sentence unclear, please reword
• Anthropometric Data
o Line 205 to 206: Change to passive voice
• Variables Value Assignment and Statistical Analysis
o Line 221 to Line 231: Move to subsection 2.3
o Line 232 to 236: Move to subsection 2.4
4) Results
• Table 1, 3, 4 and 5: Suggest using smaller font size to ease parallel viewing of all numbers
• Line 259: Remove ‘And’
• Line 345: Remove ‘The results were interesting and even counterintuitive’
5) Discussion
• The discussion section can be shortened and reorganised for better flow by focusing on the key findings on demographic, anthropometry, family food environments and food consumption
• Minor edits as follows:
o Line 354 to 356: Remove the sentence
o Line 365 to 382: Suggest to remove as repetition to Introduction
o Line 463: Change to passive voice
o Line 482: Change to passive voice
6) Grammar and spelling check
• Please ensure the manuscript is proofread and edited as there are a number of grammatical errors throughout especially on the tenses used when citing the findings from other studies, and usage of passive voice instead of active voice.
Author Response
Thank you very much for your advice and patient guidance. I've made detailed modifications.
Point 1: Abstract
- Replace ‘Objective’ with ‘Background’ to outline the context of the paper before stating the study objective.
Response: Thanks for your suggestion, it has been revised as shown below:
“Background: Family is the most fundamental and proximal context to children, and child's eat-ing behavior occurs mostly in the home or together with family members. This study aimed to evaluate the distribution of family food environment dimensions and their relations with healthy and unhealthy food consumption in primary and secondary students, to provide evi-dences in Chinese context and explore the environmental solution for children’s better nutrition.”
- Methods: Include a sentence on how results are analysed.
Response: Thanks for your suggestion, the analysis method has been included as “Binary logistic regression models were used to investigate the relationships between food consumptions frequency and FFE, and odds ratios (ORs) with 95% confidence intervals (CI) were computed for independent variables.”
- Results (Line 22): Remove ‘finally’.
Response: Thanks, the word has been deleted.
- Results (line 30): Remove ‘Especially’.
Response: The first word of the sentence has been deleted.
- Keyword: Suggest to include ‘China’.
Response: “China” has been included as keyword.
Point 2: Introduction
- The introduction section can be shortened and reorganised for better flow.
o Line 67 to 89: Suggest to combine both paragraph and discuss on family food environment, and make it succinct.
o Line 104 to 115: Combine with Paragraph 1 to provide an overview on the state of the problem globally and in China/ Beijing, and make it succinct.
Response: Thanks, the Introduction section has been reorganised succinctly as below:
There are five paragraphs to introduce the background of the study: 1) Explain the significance and importance of the study; 2) Define certain key terms of food choice and food environments; 3) Give a brief review of the relevant academic literature; 4) Identify a problem, controversy or a knowledge gap in the field of study based on literatures analysis in detail; 5) State the context and objectives of the study and hypotheses.
- Minor edits as follows:
o Line 40: Replace multiple with double/triple; Line 50 to 52: Remove the sentence; Line 53: Remove ‘actually’; Line 55: Sentence too long, suggest to stop after ‘factors’ and start new sentence.
Response: Thanks, the inappropriate words and sentence have been revised.
o Line 63 to 65: Sentence too long and unclear, please reword.
Response: The sentense has been shortened as below:
“Overall the food environments influence the opportunities of healthy eating by interacting with individual factors, which have been demonstrated as effective approaches in improving children’s diet quality and chronic disease risk.”
o Line 90 to 92: Sentence too long and unclear, please reword.
Response: The sentense has been reworded as below:
“The observational studies gave clues to explore the relationship of family factors with children’s food consumption, but there were few intervention studies to demonstrate the causality effects.”
Point 3: Materials and Methods
- Study Design and Sampling, Line 123-124: Suggest to remove sentence.
Response: The sentense has been deleted.
- Family Food Environment Assessment and Demographic Measure:
o Line 150: Change to passive voice
Response: The sentense has been modified.
o Line 163: Define ‘SDS’
Response: The “SDS” has been replaced as “family economic level”.
o Line 185 to 186: Sentence unclear, please reword
Response: The sentence has been reworded as below:
“Demographic data on the students’ grade and gender, caregivers’education, and family structure were investigated by the questionnaire either.”
- Anthropometric Data
o Line 205 to 206: Change to passive voice
Response: The sentense has been modified. And all sentences in the manuscript have been checked to modify the incorrect voice.
- Variables Value Assignment and Statistical Analysis
o Line 221 to Line 231: Move to subsection 2.3.
o Line 232 to 236: Move to subsection 2.4.
Response: The paragraphs have been moved to appropriate subsection.
Point 4: Results
- Table 1, 3, 4 and 5: Suggest using smaller font size to ease parallel viewing of all numbers
- Line 259: Remove ‘And’
- Line 345: Remove ‘The results were interesting and even counterintuitive’
Response: The font size of the tables has been reduced to fit the page space. The above word and sentence also have been deleted. Thanks.
Point 5: Discussion
- The discussion section can be shortened and reorganised for better flow by focusing on the key findings on demographic, anthropometry, family food environments and food consumption
Response: Thanks, and Discussion section has been reorganized.
- Minor edits as follows:
o Line 354 to 356: Remove the sentence.
Response: The sentence has been shortened greatly to a phrase.
o Line 365 to 382: Suggest to remove as repetition to Introduction.
Response: The paragraph has been shortened greatly and the repetition contents was deleted. The discussion about the measurement of family food environment was reserved.
o Line 463: Change to passive voice.
o Line 482: Change to passive voice.
Response: So sorry for the grammar errors, the active voice has been corrected to passive one.
Point 6: Grammar and spelling check
- Please ensure the manuscript is proofread and edited as there are a number of grammatical errors throughout especially on the tenses used when citing the findings from other studies, and usage of passive voice instead of active voice.
Response: Thanks very much. The manuscript has been checked throughly and carefully to correct English grammar and voice.
Reviewer 2 Report
This is an interesting paper on the association between family food environments (FFE) and food consumption and overweight status of primary and secondary students in Beijing, China. The most striking result that the students with healthier FFE were more likely to be overweight was appropriately discussed. The reviewer has some minor comments.
Minor comments:
1) L122
The study was conducted from September 2020 to June 2021. Please describe whether the COVID-19 pandemic had affected mobility of the students' family because a question on family travel was included in the questionnaire.
2) L203-206
It should be described how the anthropometric data were linked with the survey data of this study.
3) L163
"SDS" should be "SES".
4) L264
A total score of family food rules (FR) and family meal practices (MP) should be 24 and 17, respectively.
Author Response
Thank you very much for your advice and patient guidance. I've made detailed modifications.
Point 1: L122: The study was conducted from September 2020 to June 2021. Please describe whether the COVID-19 pandemic had affected mobility of the students' family because a question on family travel was included in the questionnaire.
Response: It is a great comment, and discussion about the item has been added in the “2.2. Family Food Environment Assessment and Demographic Measure” subsection, as shown below:
“Despite the family travel could be affected by COVID-19 pandemic, the item was kept in view of all participants living in the same context.”
Point 2: L203-206: It should be described how the anthropometric data were linked with the survey data of this study.
Response: Thanks, it has been added in the “2.4. Anthropometric Data” subsection, as shown below:
“The anthropometric data was imported in the survey database of the study by unique student number.”
Point 3: L163: "SDS" should be "SES".
Response: Thanks, the ”SDS” has been replaced with “family economic level”.
Point 4: L264: A total score of family food rules (FR) and family meal practices (MP) should be 24 and 17, respectively.
Response: Thanks for your careful review, it is our error and has been corrected.